# Effect of Nano-Ti Particles on Microstructure and Mechanical Properties of Mg-3Al-1Zn Matrix Composites

**DOI:** 10.3390/ma16062407

**Published:** 2023-03-17

**Authors:** Wei Tian, Pengfei Gao, Shengli Han, Xiaohong Chen, Fuwei Zhang, Yuhui Zhang, Tiegang Luo, Kaihong Zheng

**Affiliations:** 1School of Science, University of Shanghai for Science and Technology, Shanghai 200093, China; 2Co-Innovation Center for Energy Therapy of Tumors, Shanghai 200093, China; 3Guangdong Key Laboratory of Metal Strengthening and Toughening Technology and Application, Institute of New Materials, Guangdong Academy of Sciences, National Titanium and Rare Metal Powder Metallurgy Engineering Technology Research Center, Guangzhou 510651, China; 4School of Materials and Chemistry, University of Shanghai for Science and Technology, Shanghai 200093, China

**Keywords:** nano-Ti, Mg-3Al-1Zn matrix composite, strength, ductility, bidirectional improvement

## Abstract

In this paper, a new nanoscale metal Ti particle-reinforced Mg-3Al-1Zn matrix composite was successfully designed and prepared, which is mainly characterized by the fact that in addition to the “light” advantages of magnesium matrix composite, it also realizes bidirectional improvement of strength and ductility of the composite, and can be used as an alternative material for military light vehicle armor and individual armor. The SEM test shows that the nano-Ti particles are uniformly distributed at the grain boundary under the extruded state, which nails the grain boundary, inhibits the grain growth, and significantly refines the grain. XRD tests show that the addition of nano-Ti particles increases the crystallinity of the composite, which is consistent with the SEM test results. In addition, the EBSD test shows that the weakening of the texture of Ti/Mg-3Al-1Zn matrix composites and the increase in the starting probability of slip system are the main reasons for the improvement in ductility. Mechanical tests show that the yield strength, tensile strength, and elongation of the 0.5 wt% Ti/Mg-3Al-1Zn matrix composites exceed the peak values of ASTM B107/B107M-13 by 38.6%, 26.7%, and 20%, respectively.

## 1. Introduction

With the characteristics of mobility and flexibility, light armored vehicles undertake important tasks such as reconnaissance, command, and rescue in the complex battlefield environment [1], and the selection of their armored materials is mainly based on low density, high strength, and high ductility [2,3]. It is well known that magnesium has a low density [4], and Mehara’s research shows that the average density of medium-strength magnesium alloy is 1.78 g/cm^3^, which is converted into a specific strength roughly equivalent to 5083 aluminum armored alloy [5]. However, because magnesium matrix composites have a lower density than aluminum armor alloys, they are expected to become an alternative material for military light vehicle armor and individual armor.

However, the bottleneck of traditional magnesium matrix composite research is the mutual constraint of strength and ductility [6,7,8]. Therefore, the focus of future research on magnesium matrix composites should be around the bidirectional improvement of strength and ductility. The strength of the material is the primary factor in judging whether the material performance of military light armor is excellent, and the addition of appropriate particle reinforcement during preparation is one of the effective ways to improve the strength of the material [9,10,11,12,13]. Traditional reinforcing particles are ceramic particles such as silicon carbide [14], alumina [15], boron carbide [16], and TiC [17]. Yang‘s [18] study showed that the strength improvement of magnesium matrix composites mainly depended on the addition of reinforced particles to significantly refine grains. Wu [19] prepared SiC/AZ61 magnesium matrix composite, and the addition of SiC particles limited the growth of dynamic recrystallization grains, so that the grains were refined, and the tensile strength of the composite reached 386 MPa, while the elongation was only 4.7%. Peng [20] prepared TiB_2_/AZ31 magnesium matrix composites with TiB2 particles as reinforcement, and the tensile strength of the composite reached 237 MPa, but the elongation was only 4.6%. Habibnejad [21] prepared Al_2_O_3_/AZ31 magnesium matrix composites with Al_2_O_3_ particles as reinforcement, and the tensile strength of the composite reached 253 MPa, but the elongation was only 3.9%. Wang [22] prepared TiC/AZ91 magnesium matrix composites with TiC particles as reinforcements, and the strength of the composites was greatly improved, but the elongation was only 1.5%. It can be seen that although the addition of ceramic particles improves the strength of magnesium matrix composites, it leads to a significant loss of elongation.

Another focus of military light armor materials is the ductility of the material. The results of Ye [23] show that good interface bonding and co-deformation between the reinforcement and the matrix were the key to the improvement in the ductility of magnesium matrix composites. Zhou [24] used carbon nanotubes as reinforcement to prepare CNTs/AZ31 magnesium matrix composites; the interface of the composite materials was well bonded and its elongation reached an amazing 14%, but its yield strength was only 215 MPa. Feng [25] used Ni particles as metal reinforcement, Ni/AZ61 magnesium matrix composites were prepared, and the tensile strength of the composite reached 284 MPa, and its elongation was slightly damaged compared with the matrix. Wong [26] used Cu particles as metal reinforcement to prepare Cu/Mg composites, and its yield strength and tensile strength reached 237 MPa and 286 MPa, respectively, and its elongation was slightly damaged compared with the matrix.

Chin [27], Ho [28], Kwasniak [29], etc., show that the addition of copper (Cu), nickel (Ni), molybdenum (Mo) and titanium (Ti) can effectively improve the strength and ductility of the composite material [25,26,27,28,29,30,31,32,33,34]. However, Cu and Ni form intermetallic compounds Mg_2_Cu [35] and Mg_2_Ni [36] with Mg, which hinder the ductility of magnesium-based composites [28,35,36]. There is no interface reaction between Mo and Mg matrix [23], but its wettability with Mg and its grain refinement effect have not been widely reported. It is difficult for Ti and Mg to react with Mg to form hard or nondeformable intermetallic compounds in various metal reinforcements [37,38,39], and Ti and Mg have similar crystal structures at room temperature, which favors the interfacial binding of Ti and Mg [40,41]. In addition, Ti has good ductility [42,43,44,45], and during the deformation process, Ti will co-deform with the Mg matrix, which helps to improve the ductility of the composite. Therefore, metal Ti particles should be ideal reinforcements for the preparation of magnesium-enhanced matrix composites of metal particles.

Studies by Dieringa [30], Sankaranarayanan [46], Park [47], etc., have shown that the yield strength, tensile strength, and work hardening rate of composite materials all increase with the decrease in the particle size of the reinforcement. The decrease in particle size will lead to an increase in the strain gradient in the matrix, resulting in an increase in dislocation and an improvement in the comprehensive mechanical properties of the composite. Numerous studies have shown that the interfacial bonding between the enhancer and the Mg matrix greatly affects the final mechanical properties of the material [48,49,50], and increasing the contact area between the reinforcement and the matrix is obviously beneficial to the interfacial bonding [32]. Ye [23] prepared Ti/AZ31 magnesium matrix composites with micron Ti particles as reinforcements, and the addition of Ti particles realized the bidirectional improvement of the strength and ductility of the composites. Therefore, the design of nano-Ti-reinforced magnesium matrix composites on the basis of micron-scale reinforcements should be able to better achieve bidirectional improvement of strength and ductility.

In general, for powder metallurgy and stirred casting, the reinforcement has better dispersion in the composites prepared by powder metallurgy method [51,52]. However, the density of composites prepared by conventional powder metallurgy may have certain defects [53], and hot extrusion can improve this defect [54]. In this paper, a new nanoscale Ti particle-reinforced Mg-3Al-1Zn matrix composite was prepared by introducing a hot extrusion process based on powder metallurgy, which can be used as an alternative material for military lightweight armor.

## 2. Materials and Methods

### 2.1. Materials

In this work, spherical Ti powder with average size of 50–100 nm and purity of 99% (Jinna New Material Technology Co., Ltd., Foshan, China) and spherical AZ31 powder with average size of 100μm and purity of 99.9% were used as raw material powder (see Table 1 for elemental composition).

This study uses powder metallurgy method to prepare composite materials. The composition design of composite materials is based on the complete coating of matrix powder with reinforced body powder. The composition of the composite material is calculated as follows:(1)$$S_{1}=4\pi r^{2}$$
(2)$$S_{2}=4\pi R^{2}$$

S_1_ and S_2_ represent the surface area of Ti particles and Mg-3Al-1Zn powder, respectively. r and R represent the radio of Ti particles and Mg-3Al-1Zn powders, respectively. When the substrate powder is completely coated, it can be concluded that:(3)$$S_{2}=\frac{n}{2}S_{1}$$

In formula (3), n is the number of nano Ti particles completely coated with Mg-3Al-1Zn matrix powder. Combined with Equations (1)–(3), the following can be obtained:(4)$$n=\frac{{2R}^{2}}{r^{2}}$$
(5)$$M_{1}=\rho_{1}\frac{4}{3}\pi r^{3}$$
(6)$$M_{2}=\rho_{2}\frac{4}{3}\pi R^{3}$$

M_1_ and M_2_ represents the mass of single Ti particles and single Mg-3Al-1Zn matrix powder, respectively. $\rho_{1}$ and $\rho_{2}$ are the densities of nano-Ti particles and Mg-3Al-1Zn matrix powder, respectively ($\rho_{1}$ = 4.51 g/cm^3^, $\rho_{2}$ = 1.73 g/cm^3^). By combining Equations (4)–(6), the composition ratio of the composite material in the case of complete coating can be obtained:(7)$$M_{2}:nM_{1}=\frac{R\rho_{2}}{{2r\rho }_{1}}$$

According to the powder particle size combination Formula (7) of nano-Ti and Mg-3Al-1Zn matrix, the proportion of nano-Ti is 0.26–0.58 wt%. Therefore, the composition is designed as (0, 0.5, 1) wt% to reduce the experimental error.

### 2.2. Fabrication of the Composite

The composites were prepared using powder metallurgy subsequent hot extrusion, and the fabrication procedure is shown in Figure 1. The powder mixer SHY-5 was used to mix 0 wt%, 0.5 wt%, and 1 wt% nano-Ti with Mg-3Al-1Zn matrix powder at room temperature for 4 h. The mixed powder was loaded into a vibration tank and transferred to a vibration table model (TM2101-17) with the vibration direction of *Y* axis, a frequency of 28 Hz, and vibration for 10 min under argon gas. XGB12 planetary ball mill was used for high-energy ball milling of mixed powder in argon environment with a ball-to-material ratio of 20:1 and a speed of 250 r/min for 2 h. The homogeneously mixed composite powders were compacted into a cylinder with a diameter of 20 mm and a height of 200 mm under a pressure of 600 MPa. The Ti/Mg-3Al-1Zn-based composites were sintered at a high temperature of 120 min by heating at a heating rate of 5 °C/min in argon atmosphere at 500 °C. The composites were preheated at 400 °C for 90 min and then hot-extruded (extrusion ratio 16:1) at an extrusion speed of 0.5 m/s. The density of extruded Ti/Mg-3Al-1Zn composites was measured by digital densimeter. The theoretical density and actual density of the composites are shown in Table 2.

### 2.3. Characterization

The morphology of elemental powder and nano-Ti/Mg-3Al-1Zn mixed powder was observed by scanning electron microscopy (SEM FEI Nova Nano). The phase composition of the samples was determined by an advanced X-ray diffractometer (XRD Bruker D8). The scanning range was 2θ (20–90°) and the scanning speed was 2°/min. The microstructure of sintered nano-Ti/Mg-3Al-1Zn matrix composites and Mg-3Al-1Zn matrix was observed by light microscopy (OM Zeiss DM2500M 3.1MPCCD) and SEM. To observe the crystal structure, the sintered sample was etched using AC_2_ reagent (Table 3), and the corresponding grain size of the extruded composite and the Mg-3Al-1Zn magnesium alloy was measured.

The mechanical properties of Ti/Mg-3Al-1Zn-based composite extruded states with different mass fractions (0 wt%, 0.5 wt%, 1 wt%) were measured by performing tensile experiments at a displacement rate of 2 mm/min in the tensile testing machine (Zwiki-Z250) at room temperature. Three cylindrical samples with a diameter of 9 mm and a height of 140 mm were prepared from each component by machining for tensile test.

## 3. Results

### 3.1. Evolution of Nano-Ti/Mg-3Al-1Zn Mixed Powder during Dispersion Treatment

As shown in Figure 2a, the matrix Mg-3Al-1Zn powder used in the experiment is a regular sphere with a particle size of 100 um. The morphology of the reinforcement nano-Ti powder is shown in Figure 2b, and the particle size of the powder is 50–100 nm. As can be seen from the figure, larger aggregates composed of multiple nano-Ti particles can be observed due to van der Waals forces and electrostatic attraction between powders [55]. Figure 3 shows the scanning images of the surface of the Ti/Mg-3Al-1Zn mixed powder with different mass fraction nanoparticles. The gray spherical particles in the figure are Mg-3Al-1Zn powders and the white bright spots are nano Ti particles, and the coating rate of the substrate surface is higher with the increase in the content. In addition, larger white bright spots can be observed in Figure 3c, which are caused by the agglomeration of nano Ti particles.

### 3.2. Microstructure of Ti/Mg-3Al-1Zn Composite

Figure 4 shows the XRD patterns of the extruded state of the composites containing (0–1) wt% Ti/Mg-3Al-1Zn matrix. The figure shows the strong peaks associated with Mg, and compared with the matrix, the Mg peak gradually increases with the increase in nano-Ti content, which indicates that the addition of Ti particles increases the crystallinity of the composite and refines the grains [56]. In addition, weak Ti peaks can be observed in the XRD plots, which are not easily detected by XRD when the content of the reinforcement in the composite is less than 5 wt%.

Figure 5 shows the OM diagram and grain size statistics of the extruded Ti/Mg-3Al-1Zn matrix composites. In the composites, most of the grains are equiaxed grains rather than deformed grains, which indicates that dynamic recrystallization (DRX) occurs in the hot extrusion process [46]. The average grain size of 0 wt% Ti composites measured by EBSD is about 6.2 μm, while the average grain size of 0.5 wt% Ti composites is 4.516 μm, which is significantly reduced by 30% compared with the matrix grain size. This indicates that the addition of nano-Ti particles makes the grain of the composite material significantly refined. This shows that during the preparation process, nano-Ti particles coated on the surface of the matrix play a role in pinning grain boundaries and inhibiting grain growth. In addition, in the statistics of grain size, it is found that the grain size difference is large in the composites containing Ti, and the number of fine grains in the composites increases with the increase in Ti content, and the gap between grain sizes decreases gradually. However, the average grain size of 1 wt% Ti composite is 4.258 μm, which only decreases by 5% compared with that of 0.5 wt% Ti composite. This phenomenon reflects that the increase in nano Ti does not make the grain refinement of the composites continue. The uneven distribution of nano-Ti particles and the failure to cover all grains are the main reasons for the difference in grain size [57,58].

Figure 6 shows the SEM images of Ti/Mg-3Al-1Zn matrix composites. Figure 6a shows the scanned pattern of the Mg-3Al-1Zn matrix, which has a homogeneous composition, and no other substances were found. In Figure 6b,c, the interface of the composite material is clear, and the strip extrusion trace can be clearly observed. Further magnification of the scanning image can be observed that the gray particles are evenly distributed at the grain boundary. The EDS test results show that the gray particles are composed of Ti elements, which indicates that the nano-Ti reinforcements are uniformly distributed at the grain boundaries of the composite. This phenomenon shows that the addition of nano-Ti particles plays a role in pinning grain boundaries and inhibiting the enhancement of dislocation motion, which is effectively proof that nano-Ti particles play a dispersion-strengthening role on composite materials [59]. In addition, Figure 6c is the scanned image of 1 wt% Ti composite. From the enlarged scanning image, it can be found that there are large gray particles at the grain boundary, which are agglomerated nano-Ti particles. Due to the increase in the reinforcement content, the nano Ti powder and the matrix powder can not be completely dispersed during the mixing process, so some nano Ti particle agglomerates will be formed in the composite material [60].

Figure 7 illustrates the inverse pole figures (IPF) and pole figures of as-extruded Ti/Mg–3Al–1Zn composites from extrusion direction transverse direction (ED-TD) plane. It shows that the grain sizes of as-extruded composites decrease with the addition of Ti particles, which is accordant with the OM structure in Figure 4. The 0 wt% Ti composites possess the strongest fiber texture intensity 14.2. With the addition of Ti particles, the texture intensity of the composites gradually decreases from 14.2 to 7.58. The decreasing texture is mainly attributed to the DRX aroused by Ti particles [61].

Figure 8 shows the Schmidt factor diagram for the extruded state (0–0.5) wt% Ti/Mg-3Al-1Zn matrix composite in the TD direction, and the starting probability of the base <a>, column <a>, and cone <a> sliding system of the composite were analyzed. Figure 8a–c are the Schmidt factor statistics of the Mg-3Al-1Zn matrix composite <a> on the base surface, cylinder surface <a>, and cone surface <a>. It is not difficult to find that the base plane <a> slip system has a higher frequency between 0 and 0.35. It can be inferred that the deformation process of 0 wt% Ti composites is mainly completed through the sliding of the base plane <a>. The initiation of the base surface <a> slip system eventually leads to the formation of the (0001) <11–20> texture [62], which is also consistent with the 0 wt% Ti composite having a strong texture on the (0001) surface in Figure 7. In Figure 8b,c, the Schmidt factor of the cylinder <a> and cone <a> in the TD direction are mainly distributed in the area >0.4, which reflects that the difficulty of the slip system required to start the deformation in a certain direction of the matrix alloy is unbalanced, resulting in the difference in strength and ductility of the 0 wt% Ti composite [60,63]. As shown in Figure 8d, with the addition of nano Ti, the Schmidt factor of the sliding system of the composite substrate <a> is more widely distributed between 0 and 0.5, and the frequency decreases significantly. This phenomenon proves that the strengthening principle of nano Ti on composites is Orowan strengthening [64]. In addition, the Schmidt factor of the slip system on the cylinder <a> and the cone <a> of the composite also reflects the same situation compared with the matrix material (Figure 8e,f). It reflects the balance of the difficulty of the slip system required to initiate deformation in a certain direction during the plastic deformation process of the composite, so as to achieve a bidirectional improvement of strength and ductility [65,66,67].

Figure 9 shows the engineering stress–strain curves of sintered composites. Table 3 shows the YS (yield strength), UTS (ultimate tensile strength), and ε (elongation) values for extruded composites. The results show that with the addition of Ti particles, the YS of the composite increased from 169 MPa to 201 MPa. When Ti particles were added by 0.5 wt%, the UTS of the composite increased from 244 MPa to 304 MPa and the elongation increased from 6.1% to 8.4%. It can be measured that the addition of Ti particles increased the YS, UTS, and ε of the composite by 18.9%, 27.7%, and 40%, respectively. The strength and toughness of the 0.5 wt% Ti/Mg-3Al-1Zn matrix composites increased and reached a peak, which also matched the theoretical calculation of the composition. The improvement of the strength of the composites is mainly attributed to the dispersion-strengthening effect of nano-Ti particles. The main reason for the improvement in the strength of the composites is that the addition of nano-Ti particles nails the grain boundaries and inhibits the growth of grains during dynamic recrystallization. The improvement in the elongation of the composites is mainly attributed to the addition of nano-Ti particles, which weakens the texture of the composites and increases the probability of slip and shift start-up. In addition, the strength of the 1 wt% Ti composite is almost the same as that of the 0.5 wt% Ti composite, because the continuous addition of nano-Ti cannot continuously make the grain of the composite finer. On the contrary, the dispersion-strengthening effect of nano-Ti particles is limited by the content of the reinforcement particles, and the increase in the nano-Ti content will lead to a decrease in strength caused by the uneven distribution of the reinforcement particles. The decrease in the ductility of the 1 wt% Ti composite is also due to the agglomeration of the reinforcement particles at grain boundaries, which is consistent with the results in Figure 6c.

Table 4 shows comparative statistics on tensile strength, yield strength, and elongation of the three prepared composites and ASTM B107/B107M-13. As can be seen from the table, the yield strength, tensile strength, and elongation of the 0.5 wt% Ti/Mg-3Al-1Zn matrix composite exceed the peak value of ATSM B107/B107M-13 by 38.6%, 26.7%, and 20%, respectively, which provides options for future light armor materials. Some data of other particle-reinforced Mg-3Al-1Zn matrix composites are also shown in Table 5 [15,68,69,70,71,72]. It is known from Table 5 that SiC-reinforced AZ31 composites have relative high YS and UTS of 380 MPa and 454 MPa, respectively, while their elongation is only 4.75%. However, the graphite grain-reinforced AZ31 composite has a higher elongation of 9.0%, but both their YS and UTS are below 200 MPa. This suggests that it is difficult to bidirectional improve the strength and elongation of particle-reinforced magnesium matrix composites. Compared with other AZ31 composites, the 0.5 wt% Ti/AZ31 material in this study has higher tensile strength and better ductility, and both the strength and ductility of the composite increase, which is also consistent with the discussion in Figure 8 [73,74].

Figure 10 shows the fracture surfaces of the as-extruded composites. In Figure 10a, dimples with coarse sizes and irregular shapes appeared on the fracture surface of the 0 wt% Ti composite. It can be observed that the fracture direction in the dimple is consistent with the tensile direction, reflecting that the stress is more concentrated during deformation. In Figure 10b,c, there are two types of dimples: larger-sized dimples and a large number of small dimples. It is not difficult to observe that spherical Ti particles are adhered to the fine dimples on the surface of the fracture, indicating that the addition of nano-Ti contributes to the formation of small dimples. It is also not difficult to find that the fracture direction is not consistent with the tensile direction, which indicates that the existence of nano-Ti particles contributes to the stress dispersion during the deformation of composite materials [75]. In addition, the transverse surface can be observed at the fracture, which means that the crack passes through the Ti particles in the tensile test. The Ti particles at the fracture site are not separated from the matrix, indicating that there is a good interfacial bonding between the Ti particles and the Mg matrix. The crack propagation through Ti particles can improve the elongation of the composite [76].

## 4. Conclusions

In this paper, a novel nano-Ti-reinforced Mg-3Al-1Zn matrix composite suitable for light armor was successfully designed and synthesized, which achieved a bidirectional improvement in strength and ductility. Through XRD, SEM, EBSD, tensile tests, and other characterization or experimental methods, its microstructure and mechanical properties were studied in detail, and the following conclusions could be drawn:(1)The addition of nano-Ti particles nailed the grain boundaries, inhibited the grain growth, and significantly reduced the grain size of Ti/Mg-3Al-1Zn composites.(2)The YS and UTS of 0.5 wt% Ti/Mg-3Al-1Zn matrix composites reached 201 MPa and 304 MPa, respectively, which exceeded 38.6% and 26.7%, respectively, compared with ASTM B107/B107M-13, and the improvement of strength was mainly due to the nailing of grain boundaries and grain refinement.(3)Compared with zero-Ti alloy, the elongation of 0.5 wt% Ti/Mg-3Al-1Zn composite was significantly increased by 40%, which was 20% higher than ASTM B107/B107M-13. The improvement in ductility was the result of the weakening of the texture and the increase in the starting probability of the slip system.(4)Compared with micron Ti reinforcement, nano Ti/Mg-3Al-1Zn matrix composites have lower density and better strength and ductility, as well as meeting the requirements of mainstream light armor materials on the market for density and mechanical properties, which provides new ideas and references for the design and preparation of light armor materials with good mechanical properties.

## Figures and Tables

**Figure 1 materials-16-02407-f001:**
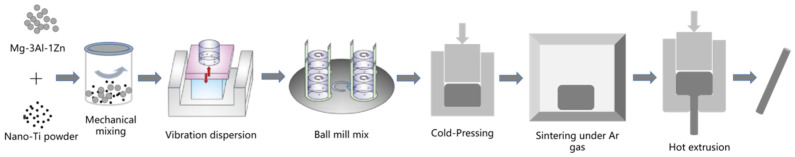
Schematic diagram of the fabrication procedure of Ti/Mg–3Al–1Zn composites.

**Figure 2 materials-16-02407-f002:**
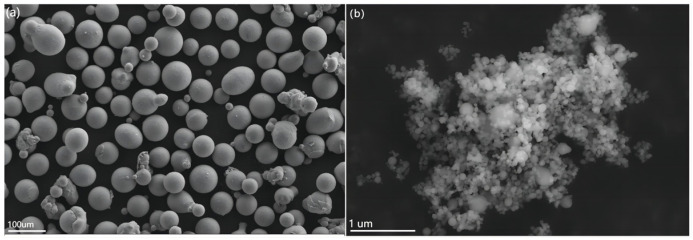
SEM view of the powder surface: (**a**) Mg-3Al-1Zn powder, (**b**) nano-Ti powder.

**Figure 3 materials-16-02407-f003:**
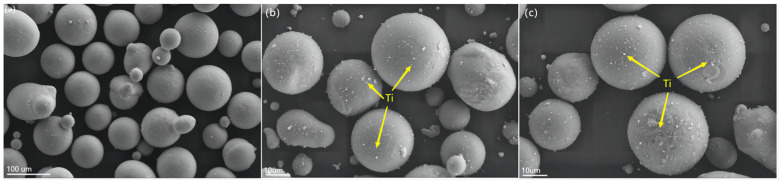
SEM view of the powder surface: (**a**) 0 wt% Ti/Mg-3Al-1Zn, (**b**) 0.5 wt% Ti/Mg-3Al-1Zn, (**c**) 1 wt% Ti/Mg-3Al-1Zn.

**Figure 4 materials-16-02407-f004:**
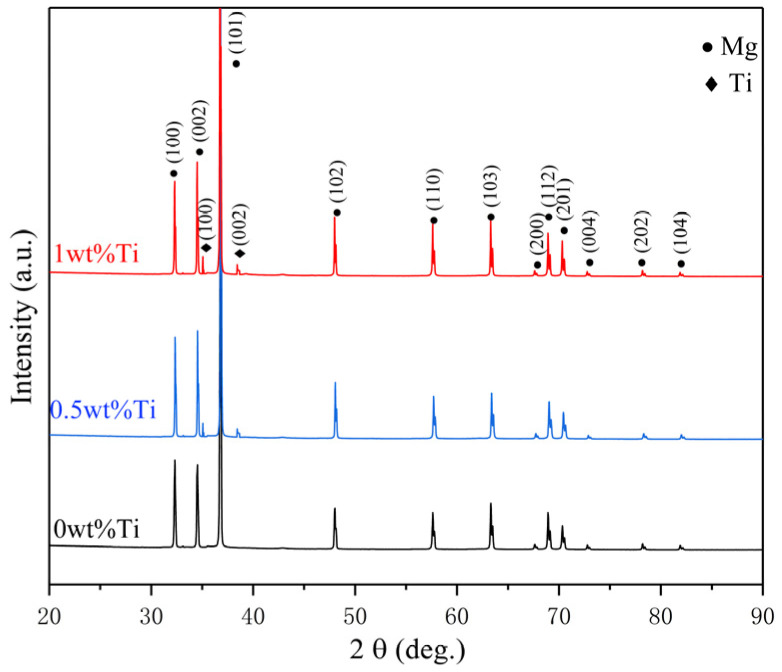
XRD patterns of as-extruded Ti/Mg–3Al–1Zn composites.

**Figure 5 materials-16-02407-f005:**
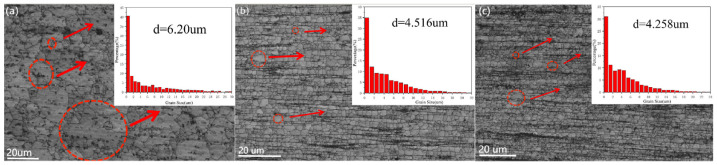
OM images of as-extruded Ti/Mg–3Al–1Zn composites: (**a**) 0 wt% Ti, (**b**) 0.5 wt% Ti, (**c**) 1 wt% Ti.

**Figure 6 materials-16-02407-f006:**
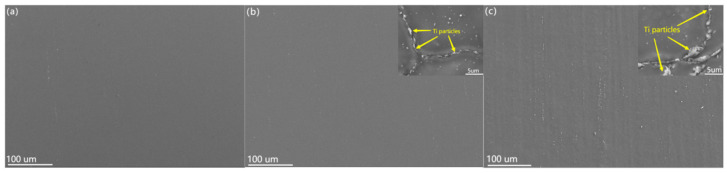
SEM images of as-extruded Ti/Mg–3Al–1Zn composites: (**a**) 0 wt% Ti, (**b**) 0.5 wt% Ti, (**c**) 1 wt% Ti.

**Figure 7 materials-16-02407-f007:**
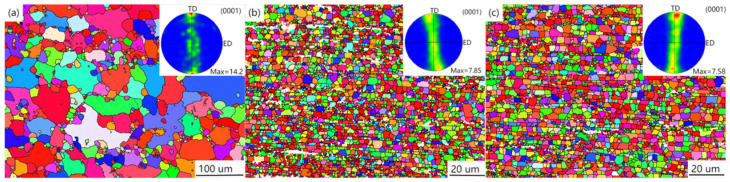
EBSD inverse pole figures and pole figures of as-extruded Ti/Mg–3Al–1Zn composites from ED-TD plane: (**a**) 0 wt% Ti, (**b**) 0.5 wt% Ti, (**c**)1 wt% Ti.

**Figure 8 materials-16-02407-f008:**
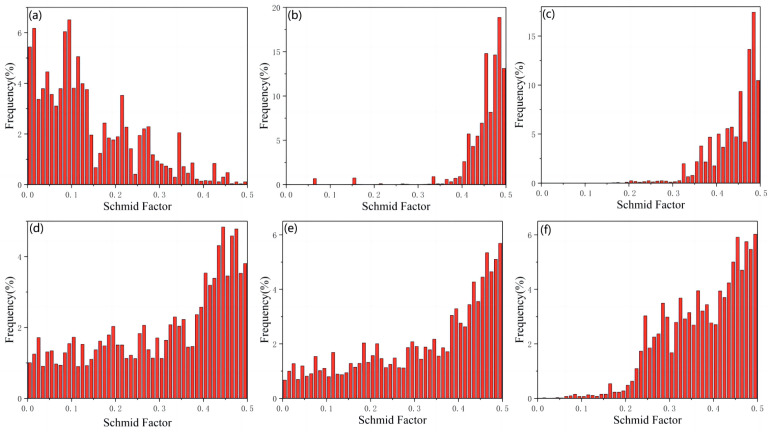
Schmidt factor plot patterns of loading in the TD direction: (**a**) 0 wt% Ti base slide, (**b**) 0 wt% Ti cylinder slide, (**c**) 0 wt% Ti pyramidal slide <a>, (**d**) 0.5 wt% Ti base slide, (**e**) 0.5 wt% Ti cylinder slide, (**f**) 0.5 wt% Ti pyramidal slide <a>.

**Figure 9 materials-16-02407-f009:**
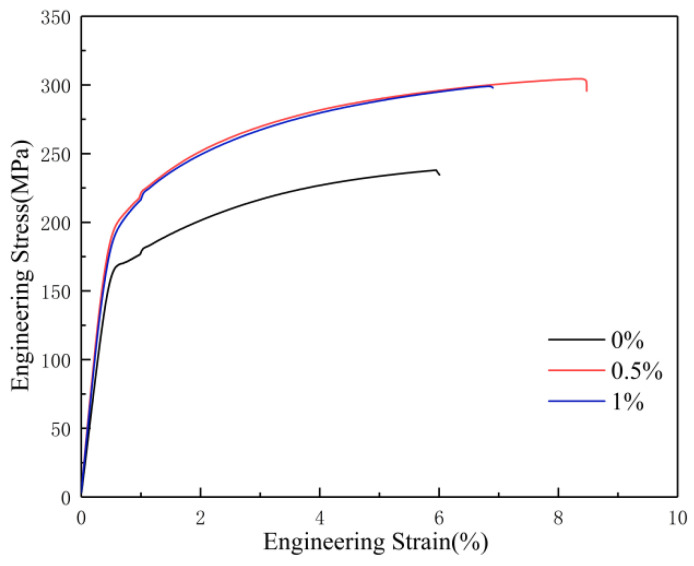
Engineering stress–strain curves of Ti/Mg-3Al-1Zn composites.

**Figure 10 materials-16-02407-f010:**
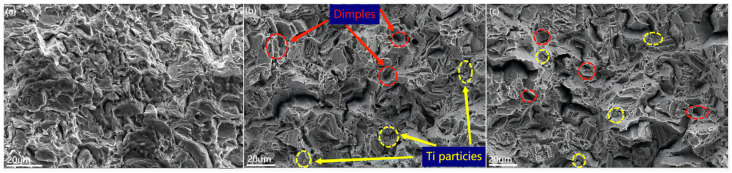
Fracture surfaces of as-extruded Ti/Mg-3Al-1Zn composites: (**a**) 0 wt% Ti, (**b**) 0.5 wt% Ti, (**c**) 1 wt% Ti.

**Table 1 materials-16-02407-t001:** AZ31 magnesium alloy.

Elemental	Mg	Al	Zn	Cu, Fe, Mn, Ni, Si
Composition (%)	Remaining	2.90	0.87	<0.005

**Table 2 materials-16-02407-t002:** Densities of the as-extruded Ti/Mg–3Al–1Zn composites.

Composite	Theoretical Density (g/cm^3^)	Actual Density (g/cm^3^)
0 wt% Ti	1.7803	1.7214
0.5 wt% Ti	1.7857	1.7425
1 wt Ti	1.7808	1.7442

**Table 3 materials-16-02407-t003:** Composition list of AC_2_ reagents.

Component	Ethanol	Propanol	Distilled Water	Perchloric Acid	Citric Acid	Hydroxyquinoline	Sodium Thiocyanate
Content	800 mL	100 mL	18.5 mL	15 mL	75 g	10 g	41.4 g

**Table 4 materials-16-02407-t004:** Stress–strain statistics of (0–1) wt% Ti/Mg-3Al-1Zncomposites and ASTM B107/B107M-13.

	YS (MPa)	UTS (MPa)	ε (%)
ASTM B107/B107M-13	145	240	7
0 wt% Ti/Mg-3Al-1Zn	169 ± 1	238 ± 1.1	6.0 ± 0.1
0.5 wt% Ti/Mg-3Al-1Zn	201 ± 1.2	304 ± 2	8.4 ± 0.5
1 wt% Ti/Mg-3Al-1Zn	194 ± 2.2	299 ± 1.5	6.9 ± 1.2

**Table 5 materials-16-02407-t005:** Tensile properties of Ti/AZ31 composites and other Mg composites.

Mg Composites	YS (MPa)	UTS (MPa)	ε (%)	Fabrication Method
0 wt% Ti	169 ± 1	238 ± 1.1	6.0 ± 0.1	PM + extrusion (this work)
0.5 wt% Ti	201 ± 1.2	304 ± 2	8.4 ± 0.5	
1 wt% Ti	194 ± 2.2	299 ± 1.5	6.9 ± 1.2	
1 wt% Al_2_O_3_ + AZ31 [70]	186	284	7.52	SPS + extrusion
9.0 wt.% Al/AZ31 [72]	166	198	7.8	Cast + FSP
7.0 vol.% Ti + AZ31 [68]	125	190	6.0	Cast + FSP
7.4 vol.% Graphite + AZ31 [69]	150	190	9.2	Cast + FSP
1 wt% SiC + AZ31 [15]	380	454	4.76	SC

Note: powder metallurgy (PM); friction stir processing (FSP); stir casting (SC).

## Data Availability

Not applicable.

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
