# Peer review of "Effect of Nano-Ti Particles on Microstructure and Mechanical Properties of Mg-3Al-1Zn Matrix Composites"

_materials, 2023, doi:10.3390/ma16062407_

Round 1

Reviewer 1 Report

The simultaneous decrease in density, an increase in strength, as well as an improvement in ductility for magnesium based alloy due to the introduction of Ti metal micro particles is shown. This looks very attractive for applications. The work makes a good impression as a materials science study, as it carefully examines the reasons for the improvement in mechanical properties. The work is well structured. I think that the material is quite worthy of publication in the journal Materials.

The simultaneous decrease in density, an increase in strength, as well as an improvement in ductility for magnesium based alloy due to the introduction of Ti metal micro particles is shown first.

I am providing some additional, specific comments below:

What are the ways to simulteneous improve in the ductilit the density and the strength for magnesium based alloy?

The only tiny remark. It seems to me that the phrase "bidirectional improvement" is unfortunate. It is better to replace it with simultaneous improvement.

Reviewer 2 Report

The author investigated the effect of Ti nanoparticles on microstructure and mechanical properties of Mg-3Al-1Zn matrix composites. The presented results are interesting and could be accepted for publication on Materials. I have some suggestions as below:

1. Provide addition information about Ti nanoparticles (commercial or lab-made). If lab-made, please add the fabrication process of Ti nanoparticles.

2. How about the hardness of the composites?

3. Due to the tensile strength of the composites are nearly the same, strengthening mechanisms of the composites should be analyzed and discussed in term of grain refinement, load transfer, dislocation density, orowan looping, etc... to explain the obtained results.

Author Response

We have included the comments on the review in the submitted PDF.

Reviewer 3 Report

Several comments should be addressed before publication. Please highlight the revision parts. The following main comments are given.

1. In "Introduction," the authors mentioned some references together, and others of that ilk. In order to manifest the relevance of references to the study, in all parts of the paper, the role of each reference should be written separately, and the results should be explained briefly. Some of the valuable recent works have not been cited. The following research papers published in the prestigious journal should be cited in the text :(doi.org/10.1016/j.jmapro.2022.11.059), (doi.org/10.1016/j.jallcom.2022.165078)

2. An updated and complete literature review should be conducted. All punctuation and sentences should be checked. The authors are recommended to seek the services of a professional copywriter or native speaker for thorough corrections. In the paper, some sentences are long, and readers cannot understand them easily. The author should write them in the standard length.

3. Please add the SEM image of the initial powders (Ti and AZ31).

4. Some pictures are not visible (Figure 4 and Figure 7), and they should be evident and large. The author should use a better color in order to show the different parts and dimensions of the figures.

5. The text, arrows, and shapes in Figure 9 are not legible. A different color should be used.

6. There should be references in the equations if they are taken from somewhere else.

7. Some of the results sections (for example lines 229-235) are reports of experimental results that should be improved with deeper discussion.

Author Response

(The authors gave the same response as above.)

Round 2

Reviewer 2 Report

The manuscript could be accepted in its current form.

Reviewer 3 Report

Authors has worked sincerely and improved the manuscripts. Therefore, I recommend to accept the paper in its present form.